# Radiosensitizing Effect of Dextran-Coated Iron Oxide Nanoparticles on Malignant Glioma Cells

**DOI:** 10.3390/ijms242015150

**Published:** 2023-10-13

**Authors:** Nhan Hau Tran, Vyacheslav Ryzhov, Andrey Volnitskiy, Dmitry Amerkanov, Fedor Pack, Aleksander M. Golubev, Alexandr Arutyunyan, Anastasiia Spitsyna, Vladimir Burdakov, Dmitry Lebedev, Andrey L. Konevega, Tatiana Shtam, Yaroslav Marchenko

**Affiliations:** 1Petersburg Nuclear Physics Institute Named by B.P. Konstantinov of National Research Centre «Kurchatov Institute», Orlova roscha 1, Gatchina 188300, Russia; nhanhau.tran92@gmail.com (N.H.T.); volnitskiy_av@pnpi.nrcki.ru (A.V.); amerkanov_da@pnpi.nrcki.ru (D.A.); pak_fa@pnpi.nrcki.ru (F.P.); golubev_am@pnpi.nrcki.ru (A.M.G.); arutyunyan_av@pnpi.nrcki.ru (A.A.); spitsyna_as@pnpi.nrcki.ru (A.S.); burdakov_vs@pnpi.nrcki.ru (V.B.); lebedev_dv@pnpi.nrcki.ru (D.L.); konevega_al@pnpi.nrcki.ru (A.L.K.); shtam_ta@pnpi.nrcki.ru (T.S.); 2Institute of Biomedical Systems and Biotechnology, Peter the Great St. Petersburg Polytechnic University, Politehnicheskaya 29, St. Petersburg 195251, Russia; 3National Research Center “Kurchatov Institute”, Akademika Kurchatova pl. 1, Moscow 123182, Russia

**Keywords:** radiosensitization, superparamagnetic iron oxide nanoparticles (SPIONs), glioma cells, radiotherapy

## Abstract

The potential of standard methods of radiation therapy is limited by the dose that can be safely delivered to the tumor, which could be too low for radical treatment. The dose efficiency can be increased by using radiosensitizers. In this study, we evaluated the sensitizing potential of biocompatible iron oxide nanoparticles coated with a dextran shell in A172 and Gl-Tr glioblastoma cells in vitro. The cells preincubated with nanoparticles for 24 h were exposed to ionizing radiation (X-ray, gamma, or proton) at doses of 0.5–6 Gy, and their viability was assessed by the Resazurin assay and by staining of the surviving cells with crystal violet. A statistically significant effect of radiosensitization by nanoparticles was observed in both cell lines when cells were exposed to 35 keV X-rays. A weak radiosensitizing effect was found only in the Gl-Tr line for the 1.2 MeV gamma irradiation and there was no radiosensitizing effect in both lines for the 200 MeV proton irradiation at the Bragg peak. A slight (ca. 10%) increase in the formation of additional reactive oxygen species after X-ray irradiation was found when nanoparticles were present. These results suggest that the nanoparticles absorbed by glioma cells can produce a significant radiosensitizing effect, probably due to the action of secondary electrons generated by the magnetite core, whereas the dextran shell of the nanoparticles used in these experiments appears to be rather stable under radiation exposure.

## 1. Introduction

Gliomas are primary tumors of the nervous system that arise from transformed neuroglial cells, accounting for about 26.6% of all brain tumors. The most aggressive and common among them is glioblastoma (adult-type diffuse glioma, IDH-wild, CNS WHO grade IV in accordance with the 2021 World Health Organization Classification of Tumors of the Central Nervous System) [1,2,3]. The standard treatment for glioblastomas involves surgical removal of tumor foci accessible for surgery, followed by irradiation and administration of the alkylating drug temozolomide. However, despite treatment, the tumor quickly recurs, leading to the death of the patient [4]. For low-grade gliomas, long-term survival exceeds 90%. The choice of therapy for this group of diseases includes careful consideration of minimizing late toxicity from surgery, chemotherapy, and radiation. Side effects of treatment may be permanent or life-threatening and include neurocognitive impairment, neurological deficits, neurovascular insufficiency, neuroendocrine insufficiency, and secondary malignancies [5].

Nuclear medicine is one of the traditional and effective areas in the fight against cancer. When radiation therapy is conducted, an important task is to ensure effective absorption of ionizing radiation in the tumor zone in order to reduce the toxic effect of radiation on the surrounding healthy tissues. Proton therapy can significantly reduce the dose load on healthy tissues [5,6]. When penetrating into matter, protons slow down losing the energy. The proton energy release peaks at the end of the proton track before the full deceleration. This feature of such a type of irradiation is called the Bragg curve, which represents the dependence of the absorbed dose on the penetration depth of the beam. The maximum of this curve is called the Bragg peak and depends on the initial proton energy [2,7,8]. After passing the maximum, the value of the absorbed dose drops sharply to zero.

Unlike protons, photon irradiation transmits energy all the way through the patient’s body, increasing initially and decreasing exponentially as photons are absorbed. Narrow proton beam magnetic scanning is used to precisely target the pathological area and limit the effect on normal tissue, a technology called intensity-modulated proton therapy (IMPT). This technology is used for both proton and photon therapy. The Bragg peak effect in proton therapy, in combination with IMPT technology, makes it possible to deliver the maximum dose to the desired area practically without affecting healthy tissues, which allows an increase in the duration and quality of life of patients [9].

To increase the therapeutic effect of radiation therapy and reduce the total dose of radiation and toxic effects on the patient’s body, further development of the methods and technologies of radiation therapy is necessary. One of the promising approaches is the development and application of combined therapies using radiosensitizers. The development and implementation of radiosensitizers into practice enables not only to reduce the total dose load and the damage of normal tissues during radiation therapy, but also increases the likelihood of suppressing highly resistant tumor cells-tumor stem cells, which are responsible for relapse of the disease after therapy.

One of the variants of radiosensitization is an introduction into the biological environment of elements with a significantly larger radiation absorption cross-section than that of the biological tissue itself [10]. The resulting short-range secondary radiation localizes the energy absorption near these elements and affects only the adjacent biological structures. This makes it possible to locally increase the absorbed dose in the target, which is determined by the distribution of the drug concentration [11]. Fe can be used as such an element in the composition of superparamagnetic iron oxide nanoparticles (SPIONs). Indeed, since the magnetic core of SPIONs has a higher density compared to cellular structures and a greater nuclear charge of Fe, the number of secondary electrons generated during the irradiation increases significantly [12]. Accordingly, the probability of damage by this short-range secondary radiation of the cell vital structures and the cell death increases when nanoparticles are absorbed by the cells. In addition, partial damage to the organic shell of SPIONs, aggravated by radiation, is possible, opening access to the magnetite core. The latter will lead to an increase in the formation of reactive oxygen species (ROS) catalyzed by the core material, which will also contribute to cell damage and death [13]. SPION radiosensitization allows achievement of a therapeutic effect at lower doses of radiation exposure, thereby reducing negative effects on healthy tissues surrounding the tumor when nanoparticles are concentrated in the tumor by targeted delivery. The considerable increase in nanoparticle concentration in gliomas was achieved with functionalized dextran-coated SPIONs [14].

The application of magnetic nanoparticles [15,16,17] opens up new opportunities in tumor theranostics: (i) it allows targeted delivery of drugs to affected areas [18]; (ii) it significantly expands the sensitivity of early diagnosis of tumors [19]; and (iii) it enables the use of non-invasive methods of treating tumors by local heating using the hyperthermia effect [20,21]. SPIONs have a number of unique properties, such as magnetism combined with the hyperthermia effect, low cytotoxicity, and high contrast for magnetic resonance imaging, which makes them suitable for targeted multimodal cancer theranostics. The therapeutic potential of SPIONs has been demonstrated in various preclinical models [11,22]. The application of SPIONs for radiosensitization further expands the possibilities for their combined multipurpose use in biomedicine.

In vitro studies of the effect of a system of nanoparticles in combination with radiation therapy on cell viability were carried out earlier in a number of works [23,24]. A cumulative effect may occur when cancer cells are exposed to ionizing radiation and agents that directly cause DNA damage and promote increased ROS production [25]. However, until now, questions about the mechanisms of radiosensitization with the use of SPIONs and their therapeutic efficacy remain open [26].

The main objective of this work was to determine the potential of dextran-coated iron oxide nanoparticles to increase the radiosensitivity of malignant glioma cells when exposed to various kinds of ionizing radiation, namely, X-rays, gamma radiation, and proton irradiation at the Bragg peak. In the case of a positive result on the radiosensitization of these particles, the second task of the work was to assess the contribution to the effect of secondary electrons formed under the action of ionizing radiation and reactive oxygen species arising from the destruction of the organic shell.

## 2. Results

Studies of the potential of iron oxide nanoparticles in a dextran shell for radiosensitization of glioma cells under the action of various types of ionizing radiation were carried out on A172 and Gl-Tr [27] glioma cell cultures (see Section 4.5).

### 2.1. Characterization of Nanoparticles

The SPIONs were synthesized as described in Section 4.1 according to a procedure that we had repeatedly tested earlier [28,29].

The SPION structure and composition were examined by X-ray diffraction (XRD) (see Section 4.2). The room temperature XRD intensity dependence on the diffraction angle is presented in Figure 1a. To evaluate the size of the nanoparticle crystallinity region, precise treatment of the XRD pattern was performed, taking into account the instrumental resolution and a doublet structure of the Cu K_λ_ line. The diffraction peaks broaden, mainly due to a finite size of the coherent scattering region and internal stress in the sample. The Williamson–Hall approach differentiates between the size-induced and strain-induced peak broadenings by considering the peak width as a function of angle [30]. Its use allowed us to evaluate the mean size of the crystallinity region ≈ 9 nm, the strain-induced broadening being small in accordance with the previous results [28].

Figure 1b displays a fragment of the transmission electron microscopy (TEM) micrograph from a glass substrate with a dried drop of a colloidal aqueous solution of SPIONs in a dextran shell (see Section 4.3). As seen in the figure, nanoparticles in the form of granules ~10 nm in diameter make up aggregates of different sizes and irregular forms. The size distribution of the aggregates (not shown) as a function of the diameter of the effective sphere approximating an aggregate is well fitted by the lognormal distribution, giving the mean value of ≈44 nm. The micrograph contrast is caused by the magnetite cores, while the dextran shells around the cores are only slightly noticeable as thin gaps (~1.5 nm) between dark spots. The mean diameter of the cores is equal within the accuracy of measurements to the XRD crystallinity size. Thus no noncrystalline phase was detected in the magnetite fraction. The particle cores are expected to be in the single-domain state, which is widely ascertained for ~10 nm magnetite particles from numerous experimental and theoretical estimations. The threshold of this state ranges from 20 to 130 nm, depending on technological conditions and other factors [31].

The average hydrodynamic radius of nanoparticle aggregates in the aqueous colloidal solution was determined by dynamic light scattering (DLS), as described in Section 4.4. Figure 1c shows that the majority of the aggregates (≈0.8 of total mass normalized to 1) had a hydrodynamic radius of about 30 nm, comparable with that from the TEM data.

### 2.2. Evaluation of the Toxicity of Nanoparticles

To establish cytotoxicity, we operated as described in Section 4.6. The results are presented in Figure 2. For both glioma cell lines, no cytotoxic effect was observed over the entire range of analyzed concentrations of SPIONs (Figure 2). For further experiments with cell irradiation, a particle concentration of 100 µg/mL was chosen.

### 2.3. Gamma Irradiation

The experimental procedure for the study of the dose dependence of gamma radiation effects on the viability of A-172 and Gl-Tr cells cultured without and with SPIONs is described in Section 4.7 and Section 4.9. The results are presented in Figure 3. The Resazurin assay (panels (a) and (b)) revealed a statistically significant difference between the viability of cells irradiated after cultivation without and with nanoparticles at each radiation dose in the range of 2–6 Gy for the Gl-Tr cell line and only at a dose of 2 Gy for A-172 cells.

The cell viability data obtained by staining with crystal violet after exposure of the cells to gamma irradiation are summarized in panels (c) and (d) of Figure 3; the data exhibit qualitative differences in the number of surviving cells cultured without/with SPIONs, corresponding to the results of the Resazurin assay. However, they are not statistically significant due to the larger error. In general, summing up the results of both tests, the sensitizing effect of SPIONs in combination with gamma irradiation is not universal and probably depends on the cell line.

### 2.4. Protons

The experimental procedure for the study of the dose dependence of proton irradiation effects on the viability of the A-172 and Gl-Tr cells cultured without and with SPIONs is described in Section 4.7 and Section 4.9. The dose dependence of cell survival of these lines, corresponding to the relative level of fluorescence in the Resazurin assay, is shown in panels (a) and (b) of Figure 4.

As can be seen from these panels, a statistically significant difference in the survival of cells incubated with or without SPIONs upon proton irradiation is observed from the Resazurin assay only for the Gl-Tr cells at the absorbed dose of 4 Gy (*p* < 0.05). Crystal violet staining, presented in Figure 4c,d, did not show a statistically significant difference in the survival of cells cultured with or without SPIONs under proton irradiation. Thus, according to the results of the two tests, nanoparticles do not exhibit a sensitizing effect for irradiation by a 200 MeV proton beam at the Bragg peak.

### 2.5. X-ray

The experimental procedure for the study of the dose dependence of X-ray irradiation effects on the viability of the A-172 and Gl-Tr cells cultured without and with SPIONs is described in Section 4.8 and Section 4.9. The results of the study are displayed in Figure 5. As can be seen from the figure, the viability of both cell lines tested by the two methods decreases at the combined employment of X-ray irradiation and nanoparticles, exhibiting a statistically significant difference from controls. This difference was observed at each radiation dose in the range 1–6 Gy.

Thus, both viability tests confirm the appearance of a combined effect upon X-ray irradiation of the cells containing nanoparticles. Figure 5e,f show photographs of plates with stained cell colonies obtained in one of the experiments, illustrating the effect of sensitization by nanoparticles.

### 2.6. Reactive Oxygen Species

To test the contribution to the radiosensitization effect from the formation of additional reactive oxygen species, we used the approach described in Section 4.10. The cell lines A-172 and Gl-Tr cultivated in the presence of nanoparticles or without them were then irradiated with a single dose of 4 Gy. The formation of ROS was determined by the intensity of fluorescence after the addition of the reagent (Fluorometric Intracellular ROS Kit) as described in Section 4. The experiment was carried out using X-ray irradiation since neither under gamma irradiation nor under irradiation with protons did the nanoparticles exhibit significant radiosensitizing properties. The dose 4 Gy was chosen on the basis that the cells should receive sufficient stress. Figure 6 shows the diagrams of the normalized ROS fluorescence level of A-172 and Gl-Tr cells cultured with or without nanoparticles after X-ray irradiation with a single dose of 4 Gy.

The average difference between ROS production by the cells cultured without/with SPIONs after X-ray irradiation is 10%, which indicates a slight increase in the level of ROS due to the combined effect of nanoparticles and irradiation. This may be evidence of the quality of the dextran shell, while the synergistic effect when using nanoparticles is obtained due to secondary electrons. At the same time, a 10% difference in the level of ROS, given that it would be a hydroxyl radical, can, to some extent, affect the result as well, leading to cell death.

## 3. Discussion

As the obtained results show, the cultivation of A-172 and Gl-Tr glioma cells with SPIONs before irradiation provides the considerable synergetic radiosensitization effect under X-ray radiation, a weak radiosensitization effect when using a gamma source, and no effect with a proton beam with the energy of 200 MeV.

The current research considers two mechanisms for the manifestation of the radiosensitizing properties of iron oxide nanoparticles: (i) an increase in the local absorbed dose due to the generation of secondary electrons by the dense core of nanoparticles; (ii) an increase in the formation of ROS, which cause oxidative stress, catalyzed by the core material when the organic shell of the nanoparticle is destroyed inside the cell [26,32,33,34]. Organically coated nanoparticles by themselves do not cause significant toxicity when absorbed by cells by means of endocytosis. Our data related to dextran-coated nanoparticles confirm this suggestion (see Figure 2). The SPIONs do not affect both studied glioma cell lines (A-172 and Gl-Tr) up to an effective Fe concentration of 400 μg/mL.

However, the nanoparticle organic shell inside the cells can be subject to decomposition when exposed to radiation due to radiation damage. This may be accompanied by an increase in the toxicity of nanoparticles. These processes must obviously depend on the type of irradiation, the size and material of the nanoparticle core, and the material and thickness of its organic shell, as well as on the cell line.

One of the pathways for the formation of reactive oxygen species in the cell is mitochondrial respiration, during which the superoxide radical O_2_^•–^ is formed. Increased formation of the superoxide radical and its conversion within the cell lead to an increase in other active forms. The superoxide radical is converted to hydrogen peroxide by the superoxide dismutase. Hydrogen peroxide in the presence of a catalyst (iron ions) is converted into a hydroxyl radical OH^•^, which has a strong reactivity and is dominant in toxic processes caused by free radicals [34,35]. Partly or completely destroyed organic shells of SPIONs inside cells under irradiation permit an interaction of the magnetite core surface with the cell cytosol and can promote the formation of ROS. Iron is present in magnetite in the form of Fe^2+^ and Fe^3+^ ions. Iron ions can be released into the cytosol and participate in the formation of a hydroxyl radical or catalyze this reaction directly on the surface of a nanoparticle when its organic shell is destroyed [13,36].

When cancer cells with incorporated nanoparticles are irradiated, the yield of secondary electrons should increase due to the higher density of the core of the nanoparticles and the greater charge of the nuclei of the metal ions (Fe) that are included in the nanoparticles. Such secondary electrons make the main contribution to cell damage [11]. Depending on the energy of ionizing radiation and the material of the nanoparticle cores, secondary electrons can be formed due to various processes: the external photoelectric effect, the Compton effect, and the formation of Auger electrons, as well as the creation of electron–positron pairs. With the external photoelectric effect, the photon energy is transferred to the electrons of the irradiated substance, after which the electrons leave the atoms of the substance, and the energy of the external quantum completely disappears. The Compton effect consists of the elastic scattering of a photon by a charged particle. Colliding with an electron, a photon changes the direction of its movement and loses some of its energy. The process of formation of electron–positron pairs occurs at energies exceeding 1 MeV—usually 2 MeV or more [13,15].

A contribution from Auger electrons can also be expected. When an atom is exposed to ionizing radiation, an electron can be knocked out of the inner shell of the atom. In place of the ejected electron, a vacancy is formed, which is occupied by one of the electrons from the outer shells, and an energy quantum is emitted, which can be absorbed by the atom itself, as a result of which the electron leaves the atom under the influence of the received energy. Such an electron is called an Auger electron. This is the case of double ionization of the atom. Such a process is typical for atoms with an electron-binding energy of no more than 1 keV. In other cases, the energy is emitted in the form of an X-ray quantum. Auger electrons have a high linear energy transfer and a short mean free path measured in nanometers. When emitters of Auger electrons are localized inside the cell nucleus, they will effectively act on the DNA double helix, leading to complex breaks. In the literature, the action of Auger electrons, when their emitters are located directly in the cell nucleus, is compared with the action of alpha particles [37,38].

In our experiments, there can be no noticeable contribution from the pair formation process since an X-ray tube with an energy of 35 keV and a source of gamma radiation with an energy of 1.2 MeV were used for irradiation. Nanoparticle cores consisting of iron oxide have a much higher density (about five times) compared to the surrounding cells and can serve as an efficient source of secondary electrons under the action of photon radiation. Consequently, at the same absorbed dose, the radiation damage to the cells loaded with nanoparticles will be greater compared to the cells without nanoparticles. As discussed above, at these photon energies, the Auger electron yield is low. When SPIONs are exposed to the 1.2 MeV radiation, the probability of photoelectric absorption is much less than that of the Compton scattering since the photon energy of the K absorption edge of Fe is 7.11 keV [39]. At low energies, such as 35 keV, on the contrary, the main contribution to the production of secondary electrons is made by photoelectric effect, and the probability of Compton scattering is substantially lower. While during X-ray irradiation, a statistically significant effect of SPION radiosensitization is observed on both cultures (Figure 5), in the gamma range, the effect is less pronounced and is present mainly in the Gl-Tr culture (Figure 3). This indicates a more efficient conversion of the energy of X-ray photons into the yield of secondary electrons on the magnetite core of SPIONs due to the photoelectric effect, which makes the main contribution to photon scattering in the region of low photon energies (10–100 keV) [12,39]. Therefore, the radiosensitizing properties of the iron oxide nanoparticles should manifest themselves mainly when cells are exposed to low-energy X-ray irradiation (Figure 5). The secondary electrons formed under ionizing radiation can damage the cell membrane or lead to a double-strand DNA break. Breaks can occur from the direct action of electrons or due to the action of free radicals formed in cells during irradiation [40]. As a comparison of Figure 3a,b indicates, according to the Resazurin assay results, the radiosensitization effect is more pronounced in the Gl-Tr culture, suggesting a specificity of cell lines to the action of gamma irradiation in the presence of SPIONs.

As mentioned above, in the presence of iron oxide nanoparticles in the cell, there is a possibility of ROS formation. The catalysis of their formation can occur during direct contact of the magnetite core of SPIONs with the cytosol after destruction of the organic shell. In the work of Klein S. et al., human colon adenocarcinoma cells (Caco-2) and human breast cancer cells (MCF-7) were incubated with uncoated or citrate- or malate-coated SPIONs and then exposed to X-rays at a single dose of 1 Gy. The irradiated Caco-2 cells incubated with citrate-coated SPIONs increased ROS concentration by 388% and, with malate-coated SPIONs, by 369% relative to the unirradiated cells. For MCF-7 cells, ROS levels after irradiation and incubation with nanoparticles increased by 327% with citrate-coated SPIONs, by 294% with malate-coated SPIONs, and by ~10% with uncoated SPIONs [41]. In our work, the increase in reactive oxygen species was only 10% for A-172 and Gl-Tr cells after cultivation in the presence of nanoparticles and photon irradiation (Figure 6). So, the rather weak effect of radiosensitization of SPIONs upon gamma irradiation of glioma cells, in addition to the lower electron yield due to Compton effect, also suggests a weak formation of additional ROS. This indicates the resistance of the dextran shell to ionizing radiation.

When irradiated with a proton beam at the Bragg peak, no statistically significant difference was observed in the viability of the A-172 and Gl-Tr cell lines cultivated in the presence of nanoparticles and without them (Figure 4). This indicates an insufficient formation of both secondary electrons and reactive oxygen species in irradiated cells. Perhaps at the Bragg peak, protons that have lost energy capture the secondary electrons knocked out by them with the formation of hydrogen atoms. Therefore, the number of secondary electrons is likely to decrease significantly, which is accompanied by a decrease in the amount of damage they cause. Probably, as in the case of photon irradiation, there is also no significant damage to the dextran shell of SPIONs and an increase in ROS generation. It is possible that ROS generation can be significantly increased by using other materials for the organic shell of nanoparticles, which would be partially or completely destroyed under the action of irradiation, exposing the surface of the nanoparticles; for example, it may be citrate or malate [41]. Here, it is important to strike a balance between the toxicity and efficiency of the SPIONs. With insufficient isolation of the surface of the nanoparticles, they will exhibit severe toxicity without irradiation. It is necessary to modify the shell of the nanoparticles so that its strength is sufficient to prevent contact of the nanoparticle core with the cytosol in the absence of irradiation and partial or complete destruction of the shell under ionizing radiation accompanied by the formation of additional ROS.

## 4. Materials and Methods

### 4.1. Synthesis of Nanoparticles

Iron oxide nanoparticles were synthesized using the method of co-precipitation of iron (III) and (II) salts in a ratio of 2:1 in an alkaline medium. An aqueous solution of ammonia was used as a precipitant. Aqueous solutions of the iron salts were placed in a reaction vessel, and then cesium chloride was added and heated to a temperature of 80 °C with vigorous stirring in the nitrogen atmosphere. When the desired temperature was reached, the precipitant was added dropwise to the reaction vessel. As a result, a black precipitate formed. After completion of the reaction, the precipitate was collected with a magnet and washed with distilled water until the pH became neutral. For colloidal stabilization, the resulting nanoparticles were coated with dextran. Iron oxide nanoparticles were mixed with the 30% aqueous solution of dextran (molecular weight 9–11 kDa) and sonicated for 30 min. During the ultrasonic treatment, the suspension was cooled in an ice bath; the temperature was controlled by a sensor (thermocouple Fluke 80BK-A/Type–K, China) directly inside the suspension and did not exceed 55(2) °C. To remove large aggregates, the suspension was centrifuged at 12,000 rpm (13,800× *g*), and the supernatant was collected. To remove excess dextran, the resulting supernatant was dialyzed against distilled water for one day in dialysis bags with a molecular weight cutoff of 14 kDa. The resulting sample was stored at 4 °C.

### 4.2. Structure Examination of Iron Oxide Nanoparticles by X-ray Diffraction

The SPION crystal structure and composition were examined by X-ray diffraction (XRD) using the DRON-3M diffractometer (Joint Stock Company “Innovation Center “Burevestnik”, Russia), providing the Cu K_α_ line radiation with the wavelength λ = 1.54 Å. To evaluate the size of the nanoparticle crystallinity region, precise treatment of the XRD pattern was performed, taking into account the instrumental resolution and a doublet structure of the Cu K_α_ line.

### 4.3. Transmission Electron Microscopy Study

A drop of the solution was put on the glass substrate, the excess water was removed with the filter paper, and the damp sample was immediately studied using the electron microscope JEM-100C (Jeol, Tokyo, Japan).

### 4.4. Size Identification of Iron Oxide Nanoparticles by Dynamic Light Scattering

The size distribution of SPIONs in the aqueous colloidal solution was evaluated by dynamic light scattering (DLS) analysis using the laser correlation spectrometers LKS-03 (INTOX MED LLC, St. Petersburg, Russia) and Photocor Compact-Z (OOO Fotokor, Moscow, Russia), as described earlier [42]. The measured autocorrelation function was fitted via nonlinear, non-negative least-square analysis. The intensity distribution was converted into the mass distribution according to the Mie theory using the supplementing software (Dynals 2.7). The measurements were carried out at 23 °C. Each sample was diluted to a concentration of 2.2 µg/mL with Milli-Q water.

### 4.5. Glioma Cell Lines and Cultivation Conditions

The study was carried out using two human glioma cell lines: the A172 cell line was obtained from the collection of the Institute of Cytology of the Russian Academy of Sciences (St. Petersburg, Russia), and the Gl-Tr cell line was generated in the Laboratory of Cell Biology (NRC «Kurchatov Institute»-PNPI, Gatchina, Russia) [27].

The complete cell growth medium contained commercial DMEM/F12 (BioLot, Hercules, CA, Russia) growth medium, 10% fetal bovine serum (Hyclone), and gentamicin. Cells were separated from the substrate using Versen’s solution. The number of the cells was estimated on a LUNA-II™ Automated Cell Counter (Logos Biosystems, Anyang-Si, Republic of Korea) after mixing the cell suspension with trypan blue (1:1).

### 4.6. Determination of SPIONs’ Cytotoxicity

To establish the cytotoxicity of SPIONs, A172 and Gl-Tr cells were plated at 2 × 10^3^ cells/well in a 24-well tissue culture plate, incubated for 24 h in complete medium, and then the medium was replaced with the medium containing SPIONs in the concentration range 0–400 µg/mL. After 24 h of incubation, the medium was replaced with the complete one, and the cells were incubated for an additional 6 days. The cytotoxic effect of SPIONs was determined using the Resazurin assay, and subsequent staining of surviving cells was performed with crystal violet. The experimental results were normalized relative to the control with a zero concentration of nanoparticles: (i) for the Resazurin assay according to the average fluorescence level and (ii) for crystal violet by the mean stained area of the control well. The experiments were carried out in triplicate.

### 4.7. Proton and Gamma Irradiation

Cells were seeded into 25 cm^2^ flasks, grown to a monolayer, and incubated with the nanoparticles at 0 and 100 μg/mL for 24 h. The cells were then removed from the flasks and transferred to the suspension containing 10^5^ cells per 1 mL of the complete medium. Then, 10^4^ cells were transferred into 0.2 mL tubes and irradiated with gamma radiation or protons. The irradiated cells were seeded into 24-well plates at 10^3^ cells per well.

Proton irradiations were performed using Roscosmos test stand IS OP CDKT.412110.013 with a 200 MeV proton beam generated by synchrocyclotron SC-1000 (NRC «Kurchatov Institute»-PNPI, Gatchina, Russia) [43]. The Spread-Out Bragg Peak (SOBP) range was 25 mm in water, and the cells were positioned in the water equivalent to the middle of the SOBP. The dose measurements were carried out at the depth corresponding to the middle of the SOBP. Proton beam calibration was performed before each irradiation. Glioma cells were irradiated with increasing doses of 0, 0.5, 1, 2, 4, and 6 Gy at the proton Bragg peak. The dose rate was 0.15–0.3 Gy/min. The experiments were carried out in triplicate.

Gamma irradiation was performed using ^60^Co γ-ray source “Issledovatel” (NRC «Kurchatov Institute»-PNPI, Gatchina, Russia) with the gamma ray energy 1, 2 MeV. Glioma cells were irradiated with increasing doses of 0, 2, 4, and 6 Gy. The dose rate was 8.5 Gy/min. The experiments were carried out in more than ten repetitions.

### 4.8. X-ray Irradiation

Cells were seeded into 24-well plates at 10^3^ cells per well. The next day, growth medium containing 100 μg/mL of nanoparticles was added to one-half of the wells, and the medium without nanoparticles was added to the other half before incubation for 24 h. Immediately before irradiation, the medium was removed from the wells, and 150 μL of fresh medium was added to prevent the cells from drying out. Cells were irradiated in 24-well plates with doses of 0, 1, 2, 3, 4, and 6 Gy. The dose rate was 5.8 Gy/min. Non-target wells were insulated with layers of lead rubber. Low-energy X-rays with the spectrum close to white noise were used. The photons emitted by X-ray tube had the energies of up to 35 KeV. The experiments were carried out in more than ten repetitions.

### 4.9. Cell Viability after Irradiation

Cell viability was determined 6 days after irradiation when the cells in the control wells almost reached a monolayer. First, the medium was taken from the wells, and then the cells were incubated with Resazurin for 1–2 h according to the manufacturer’s protocol. The Resazurin assay is based on the reduction of oxidized non-fluorescent blue Resazurin to a red fluorescent dye (resorufin) by the mitochondrial respiratory chain in live cells. The amount of resorufin produced is directly proportional to the number of living cells. The fluorescence of resorufin was detected using an EnSpire Multimode Plate Reader (PerkinElmer, Shelton, CT, USA). The wells were then washed with water and stained with crystal violet. The stained plates were photographed using the ChemiDoc system (Bio-Rad, Hercules, CA, USA), and the resulting images were processed using the free ImageJ software. The experimental results were normalized relative to the control at the zero dose of irradiation: (i) for the Resazurin assay according to the average fluorescence level and (ii) for crystal violet by the mean stained area of the control well.

### 4.10. Testing for the Appearance of Reactive Oxygen Species after Irradiation

To assess the generation of reactive oxygen species in cells after irradiation, the cells were first cultured in 96-well plates at a concentration of 10^4^ cells/well and left to grow for 24 h. After that, the cells were incubated with SPIONs (100 µg/mL) for 24 h. Next, the medium with SPIONs was removed, 100 μL of the reaction mixture (Fluorometric Intracellular ROS Kit, Sigma-Aldrich, St. Louis, MO, USA) per well was added, and incubation was carried out for 1 h. Then, one-half of the samples were irradiated with X-rays at a dose of 4 Gy. After the irradiation, the fluorescence intensity was measured, λ_ex_ = 490/λ_em_ = 525 nm, using an EnSpire Multimode Plate Reader (PerkinElmer, Shelton, CT, USA). The experimental results were normalized to the average fluorescence level of the control at zero-irradiation dose. The experiments were carried out in triplicate.

### 4.11. Statistical Analysis

Graphical visualization and statistical analysis were performed using GraphPad Prism 8.4.3. Experimental data are expressed as the mean ± the confidence interval (95% CI). To assess the differences between the groups (with or without SPIONs), the Mann–Whitney test was used. The significance level was set at *p* < 0.05.

## 5. Conclusions

The possibility of using magnetite nanoparticles coated with a dextran shell as a radiosensitizer was tested in in vitro experiments on A-172 and Gl-Tr glioma cells cultured with nanoparticles for 24 h. Three types of ionizing radiation were used: (i) 200 MeV protons at the Bragg peak; (ii) gamma radiation with a photon energy of 1.2 MeV; and (iii) X-ray radiation with the photon energy < 35 keV. The absorbed dose varied in the range 0.5–6 Gy. A statistically significant effect of radiosensitization for both types of the cell lines was observed only for the X-ray radiation, which, in this energy range, generates a significantly larger number of secondary electrons on the Fe_3_O_4_ core of nanoparticles due to photoelectric effect. These electrons lead to increased cell death. The contribution to the effect of the increase in the generation of reactive oxygen species during irradiation, as shown by the test on their formation, was small. This indicates the resistance of the dextran shell to all types of radiation in the range of the absorbed doses used. The results of the combined use of gamma radiation and SPIONs suggest a cell line-specific effect of radiosensitization, at least in this range of photon energies.

## Figures and Tables

**Figure 1 ijms-24-15150-f001:**
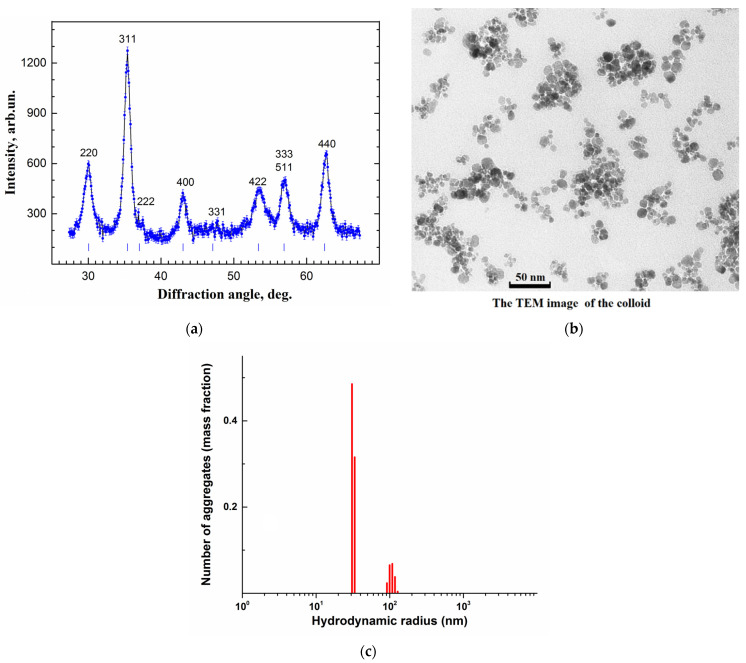
Characterization of dextran-coated superparamagnetic iron oxide nanoparticles (SPIONs). Panel (**a**) presents X-ray diffraction intensity vs. diffraction angle. The marks indicate nominal reflections for the magnetite. Panel (**b**) displays the TEM image of the nanoparticles. Panel (**c**) shows a typical example of the size distribution of the mass fractions of the synthesized dextran-coated SPIONs in the aqueous suspension measured by dynamic light scattering.

**Figure 2 ijms-24-15150-f002:**
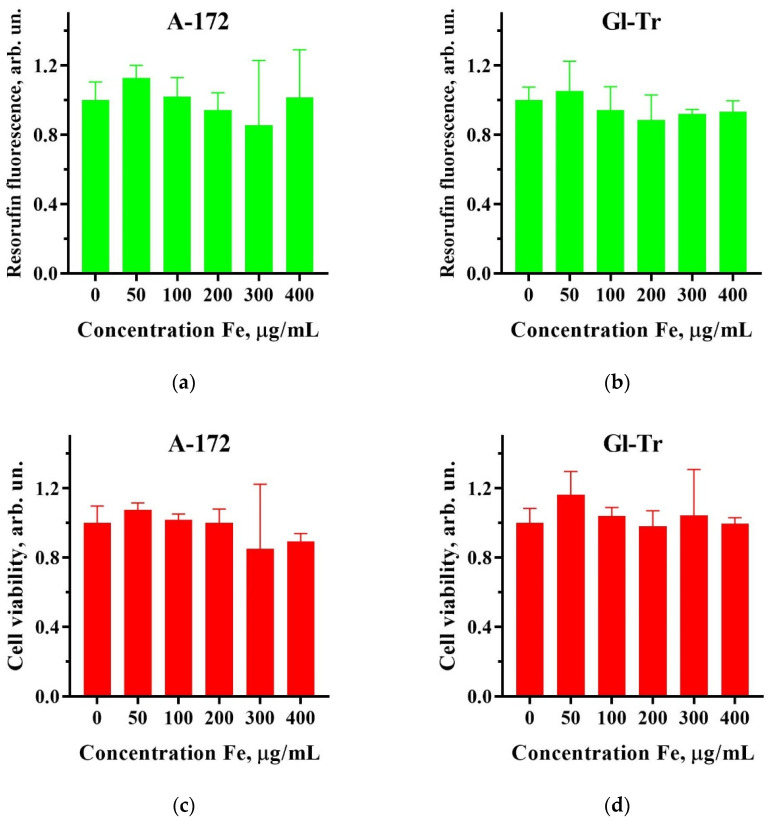
Cytotoxicity of superparamagnetic iron oxide nanoparticles (SPIONs) for A172 (**a**,**c**) and Gl-Tr (**b**,**d**) glioma cells. The cytotoxic effects of SPIONs at 0–400 µg(Fe)/mL concentration have been studied using the Resazurin assay (top panels) and by staining viable cells with crystal violet (bottom panels). Error bars are indicated for a 95% confidence level (*n* ≥ 3).

**Figure 3 ijms-24-15150-f003:**
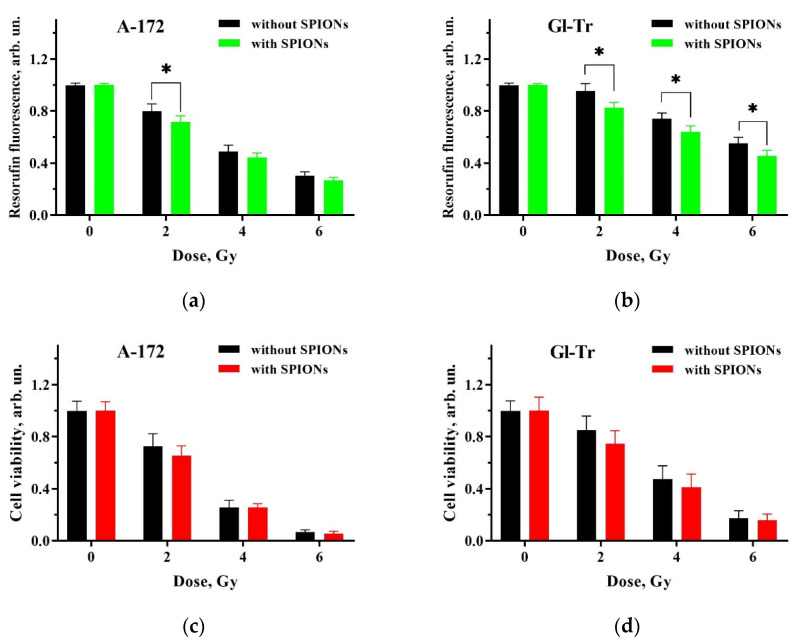
Comparison of cell survival under combined exposure to 100 µg(Fe)/mL of superparamagnetic iron oxide nanoparticles (SPIONs) and gamma radiation at different absorbed doses. Summary plots display results for the Resazurin assay (top panels) and staining viable cells with crystal violet (bottom panels). Panels (**a**,**c**) show data for the A-172 cell culture; (**b**,**d**) do the same for the Gl-Tr cell culture. Black columns refer to the cells cultivated without nanoparticles, and green or orange columns refer to the cells incubated with SPIONs for 24 h before irradiation. Errors are indicated for a 95% confidence level (*n* ≥ 10). A statistically significant difference between the results in the presence of nanoparticles and without them for the fluorescence data is indicated as * for *p* ≤ 0.001.

**Figure 4 ijms-24-15150-f004:**
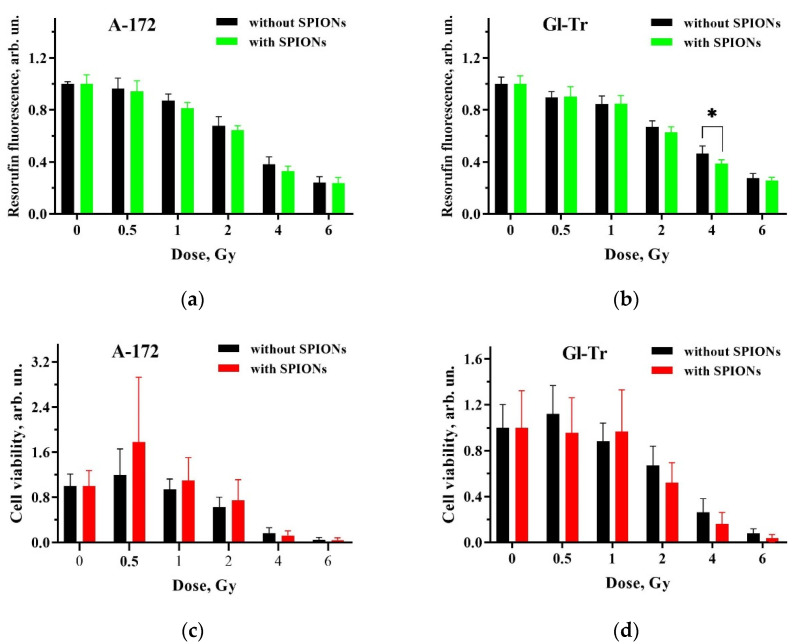
Comparison of cell survival under combined exposure to 100 µg(Fe)/mL of SPIONs and proton irradiation at the Bragg peak at different absorbed doses. Summary plots present results for the Resazurin assay (top panels) and for staining viable cells with crystal violet (bottom panels). Panels (**a**,**c**) present data for the A172 cell line, while panels (**b**,**d**) show data for the Gl-Tr cells. Black columns refer to the cells cultured without nanoparticles, and green or orange columns refer to the cells incubated with SPIONs for 24 h before irradiation. Errors are given for a 95% confidence level (*n* ≥ 3). A statistically significant difference between the results for cells incubated in the presence of nanoparticles and without them before irradiation is indicated as * at *p* < 0.05.

**Figure 5 ijms-24-15150-f005:**
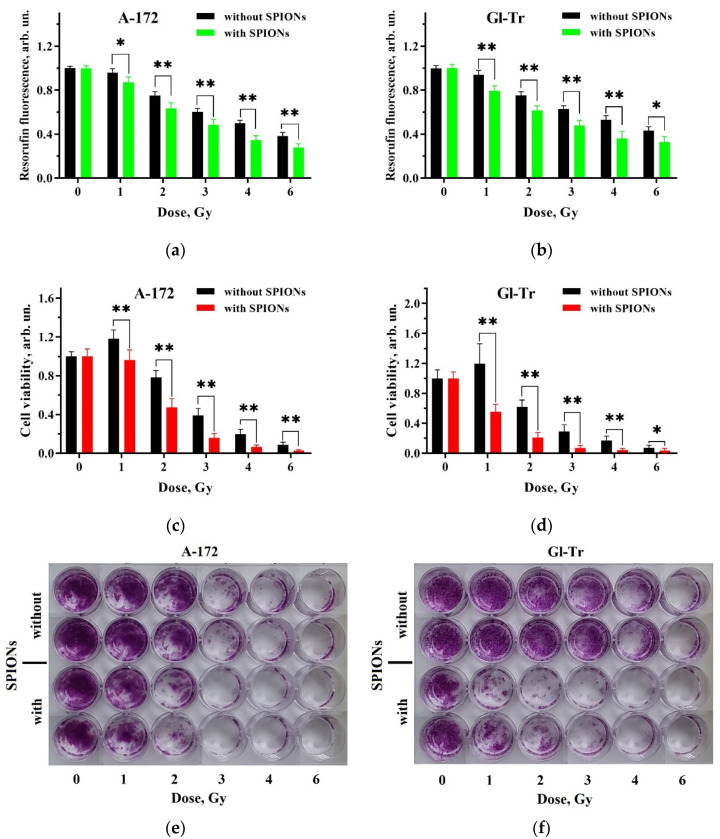
Comparison of cell survival under combined exposure to 100 µg(Fe)/mL of SPIONs and X-ray radiation at different absorbed doses. Summary plots display results for the Resazurin assay (top panels), for staining viable cells with crystal violet (middle panels), and examples of plate photographs of cells visualized using crystal violet (bottom panels). Panels (**a**,**c**,**e**) present data for the A172 cell line, while panels (**b**,**d**,**f**) show data for the Gl-Tr cells. Black columns refer to the cells cultivated without nanoparticles, and green or orange columns refer to the cells incubated with SPIONs for 24 h before irradiation. Error bars are given for a 95% confidence level (*n* ≥ 10). A statistically significant difference between the results for the cells cultivated in the presence of nanoparticles or without them is indicated as: * at *p* < 0.01 and ** at *p* < 0.001.

**Figure 6 ijms-24-15150-f006:**
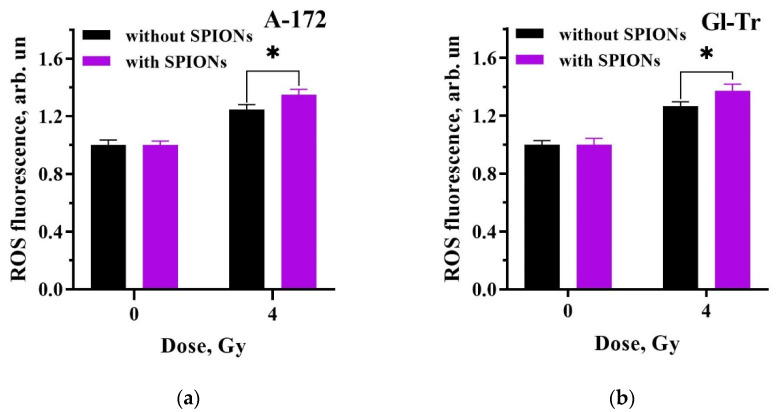
Comparison of the production of reactive oxygen species (ROS) after X-ray irradiation in cells cultured with/without SPIONs based on fluorescence intensity: (**a**) A-172 cell line; (**b**) Gl-Tr cell line. Error bars are given for a 95% confidence level (*n* ≥ 3). The statistically significant difference between the results in the presence of nanoparticles and without them is indicated as * for *p* < 0.01.

## Data Availability

The data presented in this study are available on request from the corresponding author.

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
