# Peer review of "Radiosensitizing Effect of Dextran-Coated Iron Oxide Nanoparticles on Malignant Glioma Cells"

_ijms, 2023, doi:10.3390/ijms242015150_

Round 1

Reviewer 1 Report

  IJMS (ISSN 1422-0067) ijms-2611936 Radiosensitizing effect of dextran coated iron oxide nanoparticles on malignant glioma cells

This manuscript reports on the effect of a combined radio therapeutic protocol and nanoparticles administration on glioma cell lines.

This contribution, being of a potential interest for glioma oncologists, has a quite poor scientific data in support.

Primarily, only two glioma cell lines (one of them not common in in vitro studies) have been investigated. Then, a molecular and physical characterization of nanoparticles is missing. Finally, functional studies on the effect of the combined and single therapeutic schemes is only related to viability and ROS characterization. 

I think this contribution is not suitable to be published comparing to the high scientific standards of IJMS Journal

Sincerely

Author Response

This manuscript reports on the effect of a combined radio therapeutic protocol and nanoparticles administration on glioma cell lines. This contribution, being of a potential interest for glioma oncologists, has a quite poor scientific data in support. Primarily, only two glioma cell lines (one of them not common in in vitro studies) have been investigated. Then, a molecular and physical characterization of nanoparticles is missing. Finally, functional studies on the effect of the combined and single therapeutic schemes is only related to viability and ROS characterization. 

We certainly thank the reviewer for the critical perspective on the presented research. However, we believe that exploring the potential of magnetic nanoparticles for biomedical applications is a promising area of research, as evidenced by a significant number of published works in this area. Biocompatible dextran-coated iron oxide-based nanoparticles have been investigated for potential use in cancer theranostics, both for early diagnosis and drug delivery to lesion sites, and for non-invasive therapy based on the effect of hyperthermia. The aim of this study was to test in vitro the possibility of expanding their use in cancer radiotherapy based on the sensitizing effect. In this study we used three types of ionizing radiation. Focusing our research on glioma, we used two cell lines: A172, widely used in global research practice, and Gl-Tr, which was obtained from surgical material of a patient with glioblastoma and is now being maintained as a secondary cell line, also repeatedly used in research (references 27 and 43, For example). These cell lines were chosen because they grow quickly and maintain stable parameters, which is important in the experiments. We agree that more extensive (both in vitro and in vivo) studies including different cancer types are needed to evaluate the potential of SPIONs as radiosensitizers in cancer treatment. However, we believe, that the testing of radiosensitizing effects on two cell lines is sufficient to substantiate the conclusions drawn in the study.

When studying the effect of radiosensitization by nanoparticles, it is important to monitor cell survival and ROS formation, so we did not use other approaches. It is intended to do this with the development of the work in the future.

We added X-ray diffraction (Figure 1(a)) and TEM (Figure 1(b)) data on the characterization of the synthesized iron oxide nanoparticles, and also presented these experimental results (Section 2) and discussed them (Section 3). We have also made some changes to the text of the manuscript in an effort to clarify the experimental design and make the presentation of the results clearer.

We would like to ask your attention to the fact that the study used three types of ionizing radiation. We hope that the data we obtained on the survival of glioblastoma cultures under the influence of various types of ionizing radiation will be used by other researchers when choosing a source of ionizing radiation in future work.

We hope the changes made the revised manuscript better and it will receive positive feedback.

We certainly thank the reviewer for the critical perspective on the presented research. However, we believe that exploring the potential of magnetic nanoparticles for biomedical applications is a promising area of research, as evidenced by a significant number of published works in this area. Biocompatible dextran-coated iron oxide-based nanoparticles have been investigated for potential use in cancer theranostics, both for early diagnosis and drug delivery to lesion sites, and for non-invasive therapy based on the effect of hyperthermia. The aim of this study was to test in vitro the possibility of expanding their use in cancer radiotherapy based on the sensitizing effect. In this study we used three types of ionizing radiation. Focusing our research on glioma, we used two cell lines: A172, widely used in global research practice, and Gl-Tr, which was obtained from surgical material of a patient with glioblastoma and is now being maintained as a secondary cell line, also repeatedly used in research (references 27 and 43, For example). These cell lines were chosen because they grow quickly and maintain stable parameters, which is important in the experiments. We agree that more extensive (both in vitro and in vivo) studies including different cancer types are needed to evaluate the potential of SPIONs as radiosensitizers in cancer treatment. However, we believe, that the testing of radiosensitizing effects on two cell lines is sufficient to substantiate the conclusions drawn in the study.

When studying the effect of radiosensitization by nanoparticles, it is important to monitor cell survival and ROS formation, so we did not use other approaches. It is intended to do this with the development of the work in the future.

We added X-ray diffraction (Figure 1(a)) and TEM (Figure 1(b)) data on the characterization of the synthesized iron oxide nanoparticles, and also presented these experimental results (Section 2) and discussed them (Section 3). We have also made some changes to the text of the manuscript in an effort to clarify the experimental design and make the presentation of the results clearer.

We would like to ask your attention to the fact that the study used three types of ionizing radiation. We hope that the data we obtained on the survival of glioblastoma cultures under the influence of various types of ionizing radiation will be used by other researchers when choosing a source of ionizing radiation in future work.

We hope the changes made the revised manuscript better and it will receive positive feedback.

Reviewer 2 Report

The manuscript “Radiosensitizing effect of dextran coated iron oxide nanoparticles on malignant glioma cells” by Tran, et al., demonstrates the utility of using magnetite nanoparticles coated with a dextran shell as a radiosensitizer in two glioma derived cell lines to different radiation qualities. The paper is well written. The data looks robust and was properly analyzed for significance. Similar studies have been conducted in the past by other groups using both in vitro and in vivo models. Further references and discussion should be included by the authors regarding how their findings compare to other studies. An expanded section in the discussion on the utility of the findings in vivo application would also be helpful. It would Interestingly, the authors include a comparative study of the nanoparticles across three different types of ionizing radiation with varying energies. Previous studies with dextran coated iron oxide nanoparticles demonstrated that higher dose rates were required to induce the radiosensitizing effect using higher nanoparticles’ concentration. Have the authors fully checked the literature to ensure their absorbed doses used were high enough to see an effect? Adding such experiments would increase the uniqueness and enthusiasm for these studies.  The authors also need to clearly list irradiation details such as dose rates used in the experimental methods section. The article will be of interest to general readership after major revisions. The general readership will be interested in the authors findings. I would whole heartedly support the publication of the manuscript after moderate revisions.

None

Author Response

We are grateful to reviewer for critical reading and valuable suggestions on the improvement of manuscript. The changes to the manuscript are highlighted in yellow.

The manuscript “Radiosensitizing effect of dextran coated iron oxide nanoparticles on malignant glioma cells” by Tran, et al., demonstrates the utility of using magnetite nanoparticles coated with a dextran shell as a radiosensitizer in two glioma derived cell lines to different radiation qualities. The paper is well written. The data looks robust and was properly analyzed for significance. Similar studies have been conducted in the past by other groups using both in vitro and in vivo models. Further references and discussion should be included by the authors regarding how their findings compare to other studies. An expanded section in the discussion on the utility of the findings in vivo application would also be helpful. It would Interestingly, the authors include a comparative study of the nanoparticles across three different types of ionizing radiation with varying energies. Previous studies with dextran coated iron oxide nanoparticles demonstrated that higher dose rates were required to induce the radiosensitizing effect using higher nanoparticles’ concentration. Have the authors fully checked the literature to ensure their absorbed doses used were high enough to see an effect? Adding such experiments would increase the uniqueness and enthusiasm for these studies.  The authors also need to clearly list irradiation details such as dose rates used in the experimental methods section. The article will be of interest to general readership after major revisions. The general readership will be interested in the authors findings. I would whole heartedly support the publication of the manuscript after moderate revisions.

We have tried to revise the manuscript in accordance with your recommendations.

Further references and discussion should be included by the authors regarding how their findings compare to other studies. An expanded section in the discussion on the utility of the findings in vivo application would also be helpful.

The results of the study of radiosensitization effects strongly depends of a material of nanoparticle organic cell. We did not find the results with dextran coated nanoparticles performed under close conditions for comparing with ours. We plan to use SPIONs with citrate shell in nearest future in such experiments and hope to compare new results with published in literature. Before this it seems to us too early to discuss possible utility of the findings in vivo applications.

It would Interestingly, the authors include a comparative study of the nanoparticles across three different types of ionizing radiation with varying energies.

X-ray and gamma radiation really present data for different photon energies. Unfortunately, so far we do not have sources of photon radiation with other energies. As for proton irradiation, it was carried out at the Bragg peak and the proton energy was low. We plan to increase the proton energy in the sample region, but this requires technical changes to the setup and takes time.

Previous studies with dextran coated iron oxide nanoparticles demonstrated that higher dose rates were required to induce the radiosensitizing effect using higher nanoparticles’ concentration. Have the authors fully checked the literature to ensure their absorbed doses used were high enough to see an effect?

In this work, we did not vary the dose rate during irradiation. The main goal was to obtain the dependence of the effect on the absorbed dose. The choice of the maximum absorbed dose was determined by the need for cell survival to determine the effect. At high doses, all cells (incubated/not incubated with SPIONs) died.

The authors also need to clearly list irradiation details such as dose rates used in the experimental methods section.

We have added the information on dose rates in the Materials and Methods section.

We added X-ray diffraction (Figure 1(a)) and TEM (Figure 1(b)) data on the characterization of the synthesized iron oxide nanoparticles, and also presented these experimental results (Section 2) and discussed them (Section 3). We have also made some changes to the text of the manuscript in an effort to clarify the experimental design and make the presentation of the results clearer.

Reviewer 3 Report

The study investigates the radiosensitizing effect of dextran coated iron oxide nanoparticles on  A-172 and Gl-Tr  glioma cell, evaluating in particular cell viability and ROS production. The approach and the overall design of the study are quite good. However, the authors should address the following concerns:

-The introduction could be improved in order to better clarify the presented concepts. 

-The reference and  the  relative  statement at the line 38: "The most aggressive and common among them is glioblastoma multiforme (astrocytoma grade IV)" should be changed in accordance with the 2021 World Health Organization Classification of Tumors of the Central Nervous System.

-The first part of the  "Results" section need a title, like the following parts of the same section.

-In the first part of the  "Results" section the information  about radiation doses used for cells treatment could be moved in the " Methods" section.

-In general, the discussion is not very clear and therefore could be improved.

-At the line # 266 Fe 2+ and Fe3+ need to be changed in Fe2+ and Fe3+

In my opinion, even if  I am not a native English speaker, the text is not always very comphensible and some mistakes are present.

Author Response

Reply to reviewer 3

We are grateful to reviewer for critical reading and valuable suggestions on the improvement of manuscript. The changes to the manuscript are highlighted in yellow.

The study investigates the radiosensitizing effect of dextran coated iron oxide nanoparticles on  A-172 and Gl-Tr  glioma cell, evaluating in particular cell viability and ROS production. The approach and the overall design of the study are quite good. However, the authors should address the following concerns:

We have revised the manuscript in accordance with your recommendations.

-The introduction could be improved in order to better clarify the presented concepts.”

Based on your advice, we tried to improve the Introduction by modifying the text and introducing additional references to the literature (# 3, 14, 30 in the revised manuscript).

-The reference and the relative statement at the line 38: "The most aggressive and common among them is glioblastoma multiforme (astrocytoma grade IV)" should be changed in accordance with the 2021 World Health Organization Classification of Tumors of the Central Nervous System.”

Recommended changes have been made (lines 39, 40, 41 of the revised manuscript).

“-The first part of the "Results" section needs a title, like the following parts of the same section.”

We have added a title for the first part of the Results section: 2.1. Characterization of nanoparticles. In this section we have included X-ray diffraction (Figure 1(a)) and TEM (Figure 1(b)) data in addition to DLS results for the characterization of the synthesized iron oxide nanoparticles.

-In the first part of the "Results" section the information about radiation doses used for cells treatment could be moved in the " Methods" section.”

This recommendation is fulfilled.

“-In general, the discussion is not very clear and therefore could be improved.”

Thank you very much for your helpful advice. The Discussion section have revised and, as seems to us, discussion of the results became more consistent. Changes made to the text of the manuscript are highlighted in yellow.

-At the line # 266 Fe 2+ and Fe3+ need to be changed in Fe2+ and Fe3+

Required corrections have been made, see line 296 in the revised manuscript.

Minor editing of English language required”

English language editing was carried out throughout the manuscript.

Reviewer 4 Report

This paper is expected to help researchers studying the field of treating cancer using metal oxide nanoparticles. The results of irradiating gamma, proton, and X-ray using dextran coated iron oxide nanoparticles as a photosensitizer were introduced, and the results are explained easily. However, if the dextran coated iron oxide structure and coating process are introduced, it will be easier for readers to understand. I think this paper is enough to be published in ‘International Journal of Molecular Sciences’ if only the problems shown below are solved.

1. Structure of dextran coated iron oxide

2. To confirm that the iron oxide surface is coated with dextran

3. Page 9, line 266: Fe2+ and Fe3+ --> change to superscript

Author Response

We are grateful to reviewer for critical reading and valuable suggestions on the improvement of manuscript. The changes to the manuscript are highlighted in yellow.

This paper is expected to help researchers studying the field of treating cancer using metal oxide nanoparticles. The results of irradiating gamma, proton, and X-ray using dextran coated iron oxide nanoparticles as a photosensitizer were introduced, and the results are explained easily. However, if the dextran coated iron oxide structure and coating process are introduced, it will be easier for readers to understand. I think this paper is enough to be published in ‘International Journal of Molecular Sciences’ if only the problems shown below are solved.

“However, if the dextran coated iron oxide structure and coating process are introduced, it will be easier for readers to understand.

We have made some clarifications in the description of the nanoparticle coating process in Materials and Methods section 4.1. Also, the characterization of nanoparticle ensemble state in aqueous suspension was added in Results section 2.1.

«Structure of dextran coated iron oxide»

Thank you very much for your comment, corresponding experiments have been carried out. The results are presented in the newly added Figure 1 and discussed in the Results section. We have added X-ray diffraction (Fig. 1(a), section 2.1) data of dextran coated iron oxide which confirms the magnetite structure of magnetic cores of the synthesized nanoparticles with average size ~ 10 nm. The size was estimated from the broadening of diffraction peaks due to the small size of the crystalline region. Coated nanoparticles form aggregates in aqueous suspension to reduce the magnetostatic energy of the nanoparticle ensemble due to dipole correlations of the nanoparticles in the aggregate. The average hydrodynamical radius of the nanoparticle aggregates in an aqueous colloidal solution was determined by the dynamic light scattering (DLS). DLS data (Fig. 1(c)) evidence average size of aggregate (~ 60 nm) in suspension.

«To confirm that the iron oxide surface is coated with dextran»

We have added also TEM (Fig. 1(b), section 2.1) results indicating the presence of the thin organic shell around magnetic core of the iron oxide nanoparticles. This shell is formed by dextran since at coating process of the nanoparticles a sonication was performed in a suspension of nanoparticles into a 30% aqueous solution of dextran with a molecular weight of 9-11 kDa. During ultrasonic treatment, the suspension was cooled in an ice bath (temperature not exceed 55(2)OC) preventing the destruction of dextran (section 4.1).

«Page 9, line 266: Fe2+ and Fe3+ --> change to superscript»

Required corrections have been made, see line 296 in the revised manuscript.

Round 2

Reviewer 1 Report

Dear Authors

basing on my previous criticism, your reply letter does not indicate a complete reformulation of the content of your contribution.

In particular, TEM evidence is not of a good quality with a reduced magnification. Also, and most importantly, experimental control evidence on normal cells is still missing as well as other molecular and functional studies. 

My opinion is that this manuscript is still lacking of important evidences.

Reviewer 2 Report

The manuscript “Radiosensitizing effect of dextran coated iron oxide nanoparticles on malignant glioma cells” by Tran, et al., demonstrates the utility of using magnetite nanoparticles coated with a dextran shell as a radiosensitizer in two glioma derived cell lines to different radiation qualities. The paper is well written. The data looks robust and was properly analyzed for significance. Similar studies have been conducted in the past by other groups using both in vitro and in vivo models. Adding additional  experiments in different cell lines (normal vs. cancer) would increase the impact of these studies.  The article will be of interest to general readership in its current form. I would support the publication of the manuscript in its current form.

Article needs a quick review regarding the spacing of units.